# AN EXAMINATION OF AI-GENERATED TEXT DETECTORS ACROSS MULTIPLE DOMAINS AND MODELS

**Brian Tufts**
Carnegie Mellon University
btufts@cs.cmu.edu

**Xuandong Zhao**
UC Berkeley
xuandongzhao@berkeley.edu

**Lei Li**
Carnegie Mellon University
leili@cs.cmu.edu

## ABSTRACT

The proliferation of large language models has raised growing concerns about their misuse, particularly in cases where AI-generated text is falsely attributed to human authors. Machine-generated content detectors claim to effectively identify such text under various conditions and from any language model. This paper critically evaluates these claims by assessing several popular detectors (RADAR, Wild, T5Sentinel, Fast-DetectGPT, GPTID, LogRank) on a range of domains, datasets, and models that these detectors have not previously encountered. We employ various prompting strategies to simulate adversarial attacks, demonstrating that even moderate efforts can significantly evade detection. We emphasize the importance of the true positive rate at a specific false positive rate (TPR@FPR) metric and demonstrate that these detectors perform poorly in certain settings, with TPR@.01 as low as 0%. Our findings suggest that both trained and zero-shot detectors struggle to maintain high sensitivity while achieving a reasonable true positive rate.[1]

## 1 INTRODUCTION

Large language models (LLMs) are becoming increasingly accessible and powerful, leading to numerous beneficial applications (Touvron et al., 2023; Achiam et al., 2023). However, they also pose risks if used maliciously, such as generating fake news articles or facilitating academic plagiarism (Feng et al., 2024; Zellers et al., 2019b; Perkins, 2023). The potential for misuse of LLMs has become a significant concern for major tech corporations, particularly in light of the upcoming 2024 elections. At the Munich Security Conference on February 16th, 2024, these companies pledged to combat misleading machine-generated content, acknowledging the potential of AI to deceptively influence electoral outcomes (Accord, 2024). As a result, there is a growing need to develop reliable methods for differentiating between LLM-generated and human-written content. To ensure the effectiveness and accountability of LLM detection methods, continuous evaluation of popular techniques is crucial.

Many methods have been released recently that claim to have a strong ability to detect the difference between AI-generated and human-generated texts. These detectors primarily fall into three categories: trained detectors, zero-shot detectors, and watermarking techniques (Yang et al., 2023b; Ghosal et al., 2023; Tang et al., 2023). *Trained detectors* utilize datasets of human and AI-generated texts and train a binary classification model to detect the source of a text (Zellers et al., 2019b; Hovy, 2016; Hu et al., 2023; Tian & Cui, 2023; Verma et al., 2023). *Zero-shot detection* utilizes a language model's inherent traits to identify text it generates, without explicit training for detection tasks (Gehrmann et al., 2019; Mitchell et al., 2023; Bao et al., 2024; Yang et al., 2023a; Venkatraman et al., 2023). *Watermarking* is another technique in which the model owner embeds a specific probabilistic pattern into the text to make it detectable Kirchenbauer et al. (2023). However, watermarking requires the model owner to add the signal, and its design has theoretical guarantees; we do not evaluate watermarking models in this study.

In this paper, we test the robustness of these detection methods to unseen models, data sources, and adversarial prompting. To do this, we treat all model-generated text as a black box generation. That is, none of the detectors know the source of the text or have access to the model generating the text.

---

[1]All code and data necessary to reproduce our experiments will be released publicly post-review.

This presents the most realistic scenario where the user is presented with text and wants to know if it is AI-generated or not. Specifically, we contribute:

- We conduct a thorough evaluation of AI-generated text detectors on unseen models and tasks, providing insights into their effectiveness in real-world settings.
- We analyze the performance of various detectors under adversarial prompting, exploring the extent to which prompting can be used to evade detection.
- We demonstrate that high AUROC scores, which are often used as a measure of performance in classification tasks, do not necessarily translate to practical usage for machine-generated text detection. Instead, we motivate using the metric of true positive rate (TPR) at a 1% false positive rate (FPR) threshold as a more reliable indicator of a detector's effectiveness in practice.

Due to space limitations, the discussion of related work and background is deferred to Appendix A.

## 2 BENCHMARKING PROCEDURE

Our benchmarking method involves compiling datasets that have not been encountered by any of the detectors during their training or evaluation phases. This approach ensures that the datasets represent new, unseen data and prevents the possibility of data leakage. For zero-shot detectors, this methodology eliminates the risk of using cherry-picked datasets that may bias the evaluation. For trained detectors this reduces the risk of data leakage and tests on out of domain data. Furthermore, we assess the model's performance across a diverse range of domains that the detectors may not have been previously evaluated against. This comprehensive evaluation strategy allows for a more robust assessment of the detectors' generalization capabilities. Additionally, we evaluate the detectors on a variety of language models that they have not encountered before. This approach enables us to examine the detectors' performance on unfamiliar language models, providing a more comprehensive understanding of their effectiveness and adaptability.

### 2.1 DATASETS

We evaluate each of the detectors on seven different tasks with three of the tasks, question answering, summarization, and dialogue writing, including multilingual results. The datasets chosen for each domain are as follows:

- **Question Answering:** The MFAQ dataset (De Bruyn et al., 2021) was used for this domain. It contains over one million question-answer pairs in various languages. We used the English, Spanish, French, and Chinese subsets.
- **Summarization:** We used the MTG summarization dataset (Chen et al., 2021) for this task. The complete multilingual dataset comprises roughly 200k summarizations. We utilized the English, Spanish, French, and Chinese subsets.
- **Dialogue Writing:** For this task, we utilized the MSAMSum dataset, a translated version of the SAMSum dataset(Feng et al., 2022; Gliwa et al., 2019). This dataset consists of over 16k dialogues with summaries in six languages. We utilized English, Spanish, French, and Chinese for consistency with the other multilingual domains.
- **Code:** We used the APPS dataset (Hendrycks et al., 2021), which contains 10k code questions and solutions. The subset used was randomly selected from all the data included in APPS.
- **Abstract Writing:** For this task, we utilized the Arxiv section of the scientific papers dataset (Cohan et al., 2018) to avoid potential bias, as some detectors have previously been exposed to PubMed data. Additionally, we only selected papers published in 2020 or earlier to remove potential LLM influence.
- **Review Writing:** The PeerRead dataset was used for the review writing task (Kang et al., 2018). PeerRead contains over 10k peer reiviews written by experts corresponding to the paper that they were written for.
- **Translation:** We used the Par3 dataset (Karpinska et al., 2022), which provides paragraph level translations from public-domain foreign language novels. Each paragraph includes at least 2 human translations of which we selected only one to represent human translation.

## 2.2 LARGE LANGUAGE MODELS

Our objective is to evaluate the detectors on models they they have not previously been trained or assessed on to gauge their generalization capabilities. We evaluated 4 different models across every task. The models we use are Llama-3-Instruct 8B (AI@Meta, 2024), Mistral-Instruct-v0.3 (Jiang et al., 2023), Phi-3-Mini-Instruct 4k (Abdin et al., 2024), and GPT-4o.

## 2.3 DETECTION MODELS

The detection models were chosen from the newest and highest performing detectors in their respective categories. Our goal was to represent both trained and zero-shot detectors. As previously mentioned, the trained detectors we are using are RADAR (Hu et al., 2023), Detection in the Wild (Wild) (Li et al., 2024), and T5Sentinel (Chen et al., 2023). The zero-shot detectors we are using are Fast-DetectGPT (Bao et al., 2024), GPTID (Tulchinskii et al., 2024), and LogRank (Mitchell et al., 2023).

Notably, we did not include any watermark detectors. The primary reason for this is that the evaluation techniques we use over various models would not work with watermark detection. While watermark detection has shown strong performance (Kirchenbauer et al., 2023), they have a significant drawback in that they only work if a model applies a watermark. In this paper, we assume a scenario in which no watermark is applied or it is unknown whether a watermark is applied. Therefore, we must turn to other detection methods.

## 2.4 EVALUATION METRICS

In this study, we evaluate machine-generated text detectors using AUROC and TPR at a fixed FPR. Our findings, consistent with prior research (Krishna et al., 2024; Yang et al., 2023a), suggest that AUROC alone may not reflect a detector's practical effectiveness, as a high AUROC score can still correspond to significant false positive rates. This is critical since false positives, particularly in fields like academia and media, can have severe consequences. We argue that TPR at a given FPR should be the standard evaluation metric, as demonstrated by a detector achieving a 0.89 AUROC but less than 20% TPR at a 1% FPR on a task.

## 2.5 RED TEAMING

We employ two different methods of prompting for every task: plain prompting and adversarial prompting. Plain prompting involves using a typical assistant system prompt and providing the model with the same input that was given to the human for human-generated content. Adversarial prompting, on the other hand, requests that the model try to act more like a person. Examples of the question answering plain and adversarial prompts[2] are shown as follows:

> **Plain Prompt Example: Question Answering**
>
> You are a helfpul question answering assistant that will answer a single quesetion as completely as possible given the information in the question. Do NOT using any markdown, bullet, or numbered list formatting. The assistant will use ONLY paragraph formatting. **Respond only in {language}**.

> **Adversarial Prompt Example: Question Answering**
>
> {Question answering prompt} Try to sound as human as possible.

We also conducted experiments using the LLMs as writing assistants. Specifically, we requested that the model rewrite the human response and improve upon its clarity and professionalism. This represents a scenario where a person will write down an answer first and then request that a model make their answer better before presenting it. The specific prompt we used it as follow:

---

[2]The others can be found in the appendix Table 12.

| Task | AI Avg | AI Min | Human Avg |
|------|--------|--------|-----------|
| Code | 502.16 | 15 | 4496.88 |
| QA | 508.19 | 19 | 1052.37 |
| Summ | 411.57 | 16 | 191.00 |
| Dialogue | 378.26 | 15 | 402.13 |
| Reviews | 549.51 | 24 | 796.06 |
| Abstract | 425.89 | 32 | 2081.88 |
| Translation | 525.43 | 9 | 772.75 |

Table 1: Average and minimum token counts of machine-generated and human-generated text for each task, tokenized using the Llama2-13B tokenizer (Touvron et al., 2023). Minimum token counts for human-generated text are omitted as they were previously described.

> **Rewriting Prompt**
>
> You are a helpful writing assistant. Rewrite the following text to improve clarity and professionalism. Do not provide any other text. Only provide the rewritten text.

## 3 EXPERIMENT

**Dataset Processing.** Each dataset undergoes additional processing to prepare it for detection tasks. Research indicates that detectors of machine-generated text are more effective with longer content (Yang et al., 2023b). To leverage this, we aimed to use human samples of maximum possible length. However, the minimum length needed to obtain sufficient samples varied by task. We randomly selected 500 samples of human text from filtered subsets with the following lengths: 500 tokens for question answering, 400 tokens for code[3], 150 tokens for summarization, 275 tokens for dialogue, 500 tokens for reviews, 500 tokens for abstracts, and 500 tokens for translation (Table 1). These 500 samples served as human examples. From them, prompts from the first 100 samples were chosen for use in the generator model, using the input given to the human author as the model prompt. This resulted in a dataset of 500 human examples and 100 machine-generated examples per model for a total of 400 machine-generated examples for each task. This slight data imbalance is intentional to ensure a more accurate TPR@FPR metric.

Detection methods show improved performance with longer text sequences (Wu et al., 2023) so we show the statistics of the text in Table 1. Our primary focus was on detectors' ability to identify AI-generated text while maintaining a low FPR. The longer length of human-generated text is likely to enhance the TPR@FPR by making it easier to detect as human. We considered the AI-generated text sufficiently long for two reasons. First, Li et al. (2024) reports an average AI generation length of 279.99, which is much lower than our average token lengths. Their extensive training and evaluation data support the adequacy of this length for AI content. Second, our models, with a maximum generation length of 512 tokens [4], produced responses indicative of real-world lengths.

---

[3]Length limited to 2500 tokens

[4]The averages can exceed this number due to different tokenizers and additional tokens to keep text coherent

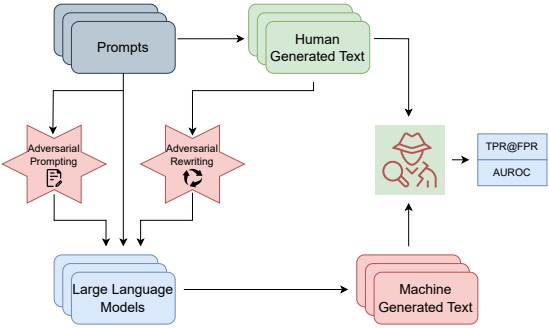

Figure 1: Pipeline for prompting and evaluation. Adversarial prompting and rewriting are applied to the LLMs. After collecting machine-generated text, AUROC and TPR@FPR are measured for each detector.

| Detector | TPR@0.01 | TPR@0.05 | TPR@0.1 | AUROC |
|---|---|---|---|---|
| Radar | 0.06 | 0.17 | 0.29 | 0.6085 |
| Fast-DetectGPT | 0.48 | 0.60 | 0.67 | 0.8376 |
| Wild | 0.13 | 0.20 | 0.30 | 0.6888 |
| PHD | 0.00 | 0.02 | 0.05 | 0.3204 |
| LogRank | 0.00 | 0.01 | 0.02 | 0.2235 |
| T5Sentinel | 0.02 | 0.06 | 0.12 | 0.4798 |

Table 2: Performance of different detectors across the entire dataset

**Text Generation and Detection Process.** Once the prompt samples were selected, we needed to generate positive examples. The process for this can be seen in Figure 1. We employ three different strategies for prompting the models. The first strategy involves using a basic prompt for each domain that explains the goal of the model and the desired output format. The second strategy consists of requesting that the model be as human as possible. The third strategy requests that the model rewrite and improve upon the human written response [5]. The first strategy aims to simulate a basic system prompt that would generally be in place on a model someone is using to generate content. The second strategy simulates the case where a user might try to get the model to generate content that closely resembles human-generated content. The third strategy simulates a scenario where the user writes their own response and simply wants the model to clean it up or make it easier to understand. The outputs of the models were taken as is with no editing. After generating the positive examples, we passed all of the machine-generated and human-generated examples through the detectors. RADAR, Fastdetectgpt, Wild, and T5Sentinel all return a percentage probability for each class, and GPTID and LogRank return a value representing their score. We do not use any thresholds and take the scores as is for AUROC and TPR@FPR metrics.

## 4 RESULTS AND ANALYSIS

Table 2 shows the overall performance of each detector across the entire dataset. In this section, we break down the performance of each detector across tasks, languages, and prompt techniques.

### 4.1 PLAIN PROMPTING

We evaluate the AUROC and TPR at 0.01 FPR for machine-generated texts from direct prompting using identical prompts as human written texts. A simple prompt was employed to ensure the generated text was in the correct format and language for the multilingual tasks.

Figures 2a and 2b show the results for the multilingual tasks and 3a and 3b show the results for the only English tasks. A significant difference is observed in detector performance across languages and tasks, particularly in the multilingual setting. Fastdetectgpt consistently performs well overall but encounters challenges in summarization tasks, especially in languages other than English. Other

---
[5]Prompts and templates in appendix

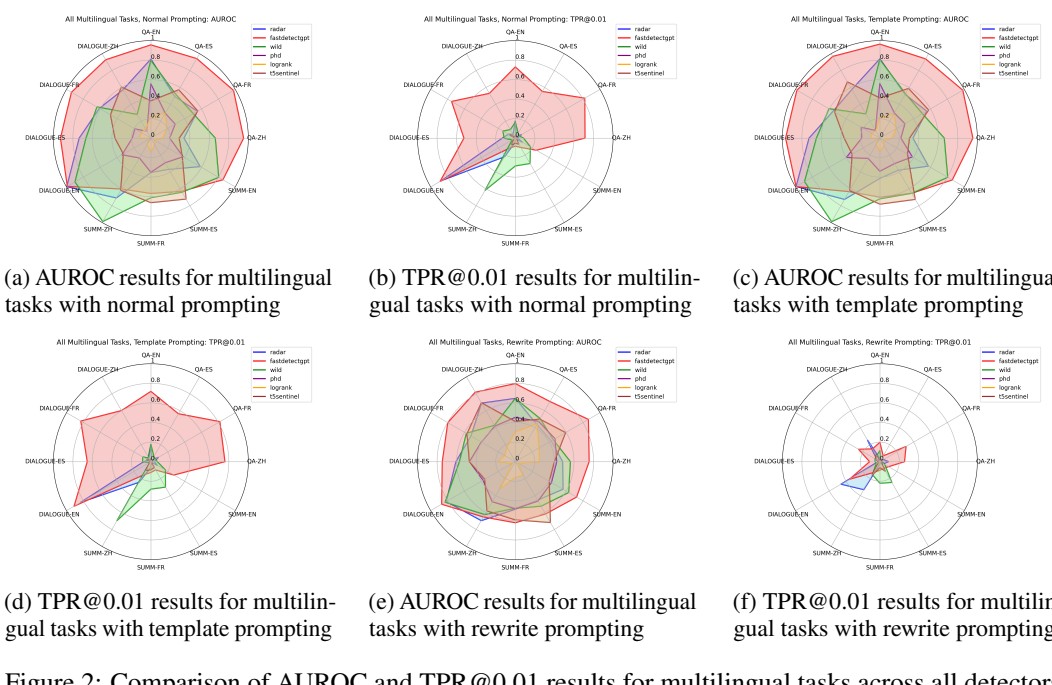

(a) AUROC results for multilingual tasks with normal prompting

(b) TPR@0.01 results for multilingual tasks with normal prompting

(c) AUROC results for multilingual tasks with template prompting

(d) TPR@0.01 results for multilingual tasks with template prompting

(e) AUROC results for multilingual tasks with rewrite prompting

(f) TPR@0.01 results for multilingual tasks with rewrite prompting

Figure 2: Comparison of AUROC and TPR@0.01 results for multilingual tasks across all detectors using different prompting styles (normal, template, and rewrite).

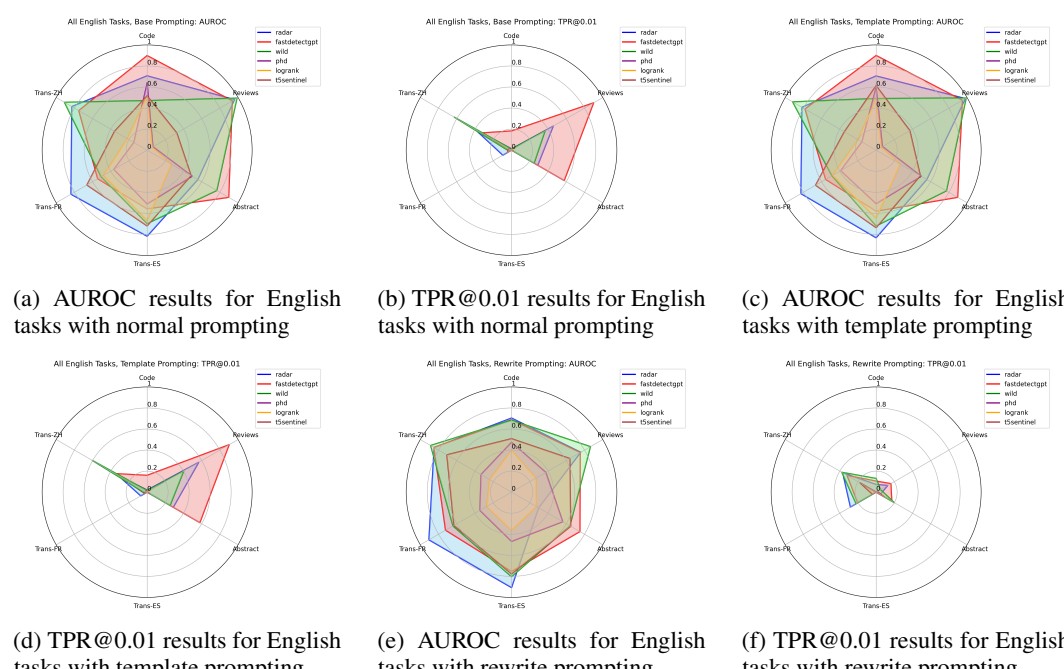

(a) AUROC results for English tasks with normal prompting

(b) TPR@0.01 results for English tasks with normal prompting

(c) AUROC results for English tasks with template prompting

(d) TPR@0.01 results for English tasks with template prompting

(e) AUROC results for English tasks with rewrite prompting

(f) TPR@0.01 results for English tasks with rewrite prompting

Figure 3: Comparison of AUROC and TPR@0.01 results for English tasks across all detectors using different prompting styles (normal, template, and rewrite).

detectors show similar patterns: while they achieve strong results in English tasks, their performance becomes more inconsistent with non-English tasks. The AUROC graph suggests robust performance for Fastdetectgpt, but when examining the TPR@0.01 graph, we observe that it struggles to maintain low false positive rates, particularly in summarization tasks where it falls below 0.25 for most languages, except for English dialogue and French question-answering, where it exceeds 0.8.

For the English-only tasks, most detectors show improved performance. In these tasks, Wild and Radar demonstrate competitive performance with Fastdetectgpt, which struggles in the translation domain. Despite expectations that the translation domain would be the most challenging due to lower entropy in translated texts, detectors performed reasonably well. While the AUROC graph indicates promise, the TPR@0.01 graph highlights ongoing challenges in maintaining low false positive rates. Additionally, while Radar and Wild outperform Fastdetectgpt in the review domain based on AUROC, they fall short in TPR@0.01 compared to Fastdetectgpt's performance.

### 4.2 TEMPLATE PROMPTING

Figures 2c and 2d show the results on the multilingual tasks where the model was instructed to be "as human as possible." Interestingly, this request had little effect on performance. In the few instances where changes occurred, scores generally increased, suggesting that asking the model to "sound human" may have made its output easier to detect. This aligns with expectations, as large language models are already trained on predominantly human-written texts, and generating more conversational output can make detection more straightforward, as evidenced in dialogue generation tasks.

On the English tasks, as shown in figures 3c and 3d, the results were similarly unaffected by the human-like request, with some slight score increases where changes were observed. This is especially expected in domains such as reviews, code, and abstracts, which follow specific writing conventions, while tasks like question answering and dialogue generation exhibit more variability and creativity.

### 4.3 REWRITING

Finally, we show the results for the rewriting prompt for the multilingual tasks in figure 2e and 2f and for the English tasks in figures 3e and 3f. We observe a notable decrease in AUROC performance for detectors that previously performed well, such as Fastdetectgpt, Radar, and Wild, while Phd and LogRank see an increase in performance, with T5Sentinel remaining largely unaffected. This performance decline is even more pronounced in TPR@0.01, where none of the detectors show improvement. Despite these shifts, the relative performance across tasks remains consistent, indicating an inherent variability in detectability based on the type of task and language.

### 4.4 TPR@FPR VS AUROC

In this paper, we utilize both the AUROC and TPR@FPR metrics. However, we also argue that TPR at a low FPR is a much more important metric for this detection task. Figure 4 shows the correlation between TRP scores at various FPR rates and the AUROC score for all tasks, detectors, and models used in this research. The AUROC correlates much higher with FPR rates in the 0.4 to 0.6 range and much lower with FPR rates at the edges, less than 0.2 and greater than 0.8. While the 0.75 is still a reasonable correlation value, the AUROC is still much more representative of the middle FPR's while we are really concerned with the lower FPR's for this task. This is why we report the TPR@0.01, which is much more representative of the applicability of a detector than the AUROC.

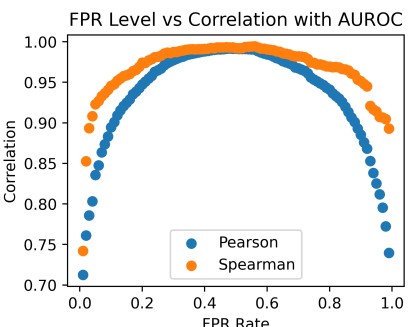

Figure 4: Correlations between various FPR rates and the overall AUROC score. AUROC score is much more representative of the middle FPR rates, while this detection task is much more concerned with the lower end of FPR.

### 4.5 OUTPUT QUALITY AND DETECTION

Measuring the quality of LLM outputs, especially in creative tasks, remains challenging, making it difficult to determine if higher-quality outputs are harder to detect. Table 3 compares various models' performance scores and rankings from Chatbot Arena (Chiang et al., 2024), allowing us to explore if output quality affects detectability. The data shows little difference in detectability across models of

| Detector | Code | | Reviews | | Abstract | | QA | | Summ | | Dialogue | | Trans. | | Arena Score |
|---|---|---|---|---|---|---|---|---|---|---|---|---|---|---|---|
| | TPR | AUC | TPR | AUC | TPR | AUC | TPR | AUC | TPR | AUC | TPR | AUC | TPR | AUC | |
| **GPT-4o** | 0.05 | 0.56 | 0.00 | 0.64 | 0.05 | 0.55 | 0.01 | 0.54 | 0.00 | 0.49 | 0.00 | 0.55 | 0.00 | 0.52 | 1339 |
| **Llama-3** | 0.01 | 0.53 | 0.00 | 0.60 | 0.01 | 0.58 | 0.00 | 0.57 | 0.00 | 0.53 | 0.00 | 0.56 | 0.00 | 0.55 | 1152 |
| **Mistral** | 0.04 | 0.56 | 0.00 | 0.63 | 0.02 | 0.52 | 0.00 | 0.57 | 0.00 | 0.50 | 0.01 | 0.57 | 0.00 | 0.54 | 1072 |
| **Phi-3** | 0.04 | 0.57 | 0.00 | 0.63 | 0.01 | 0.59 | 0.00 | 0.51 | 0.00 | 0.52 | 0.00 | 0.55 | 0.00 | 0.56 | 1066 |

Table 3: Model performance (AUROC and TPR@0.01) across tasks compared with model generation quality. The Chatbot Arena score is utilized to measure the quality of a model. The higher scores do not correlate with lower detectability of generated content.

varying quality, with AUROC and TPR@0.01 scores remaining consistent. This suggests that output quality does not significantly impact the difficulty of detection, though further research is needed for a fuller understanding.

## 5 CONCLUSION

This study evaluates six advanced detectors across seven tasks and four languages, revealing notable inconsistencies in their detection capabilities. We also examined three different prompting strategies and their impact on detectability, finding that requests for more "human-like" output do not make the text harder to detect, while rewritten human content proves more difficult to identify.

Additionally, this research highlights the limitations of relying on the AUROC metric for assessing machine-generated content detectors. Our findings emphasize the need for robust evaluation methods to develop more reliable detection techniques. The study underscores the challenges in detecting machine-generated text, particularly when human written text was only modified by a language model, and advocates for TPR@FPR as the preferred evaluation metric to better capture detector performance.

## 6 LIMITATIONS

A limitation of this method is the settings in which the human data was collected may vary from the settings in which these detectors will be used. Additionally, some of the datasets we used had collected their data from the internet which raises a concern that some of that data is not completely human generated. This is a challenge that all future detectors will also struggle with when training and evaluating. These results pose the risk of emboldening users to use AI generated content when they otherwise should not because they know detectors cannot be confidently trusted. However, acknowledging this is important to encouraging research into new detection methods and improving current methods.

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

## A    RELATED WORK AND BACKGROUND

There is a variety of related work that discusses text detectors. These works cover different aspects, such as the text detectors themselves, their types, evaluation, and red-teaming of detectors.

**Text Detectors.** Machine-generated text detectors can be divided into trained classifiers, zero-shot classifiers, and watermark methods (Yang et al., 2023b; Hans et al., 2024; Ghosal et al., 2023; Jawahar et al., 2020). (1) Trained detectors use classification models to determine if the text is machine-generated or human-written (Zellers et al., 2019b; Hovy, 2016; Hu et al., 2023; Tian & Cui, 2023; Verma et al., 2023). However, the increasing prevalence of machine-generated content (European-Union, 2022) makes it difficult to label human-generated work for training, as even humans find it hard to distinguish between the two (Darda et al., 2023). (2) Zero-shot detectors leverage intrinsic statistical differences between machine-generated and human-generated text (Gehrmann et al., 2019; Mitchell et al., 2023; Bao et al., 2024; Yang et al., 2023a; Venkatraman et al., 2023). Proposed methods include using entropy (Lavergne et al., 2008), log probability (Solaiman et al., 2019), and more recently, intrinsic dimensionality (Tulchinskii et al., 2024). (3) Watermark-based detection, introduced by Kirchenbauer et al. (2023), involves embedding a hidden but detectable pattern in the generated output. Various enhancements to this method have been suggested (e.g., (Zhao et al., 2023; Lee et al., 2023)). This paper focuses on the black-box setting, which closely resembles real-world detection scenarios. Watermarking is not tested due to its guaranteed detectability and low false positive rates (e.g., (Zhao et al., 2023)). The primary concern is detecting un-watermarked text, as it is the most commonly encountered and poses the greatest threat.

**Evaluation of Text Detectors.** The most commonly utilized metric in evaluating detectors is the area under the receiver operating curve (AUROC) (Mitchell et al., 2023; Sadasivan et al., 2023). Although it offers a reasonable estimate of detector performance, research by Krishna et al. (2024); Yang et al. (2023a), and our experimental results demonstrate that there can be a substantial difference in performance between two models with AUROC values nearing the maximum of 1.0. Consequently, the true positive rate at a fixed false positive rate (TPR@FPR) presents a more accurate representation of a detector's practical effectiveness.

**Redteaming Language Model Detectors.** AI text detectors are increasingly evaluated in red teaming scenarios, with recent contributions from Zhu et al. (2023); Chakraborty et al. (2023); Kumarage et al. (2023); Shi et al. (2024); Wang et al. (2024). Shi et al. (2024) identifies two main evasion techniques: word substitution and instructional prompts. Word substitution includes query-based methods, which iteratively select low detection score substitutions, and query-free methods, which use random substitutions. Instructional prompts, akin to jailbreaking, instruct the model to mimic a human-written sample. Query-based word substitution proved most effective, reducing the True Positive Rate (TPR) to less than 5% at a 40% False Positive Rate (FPR) against DetectGPT.

Wang et al. (2024) explores robustness testing with three editing attacks: typo insertion, homo-glyph alteration, and format character editing. Typo insertion adds typos, homoglyph alteration replaces characters with similar shapes, and format character editing uses invisible text disruptions. Paraphrasing attacks, noted by Krishna et al. (2024), include synonym substitution (model-free and model-assisted), span perturbations (masking and refilling random spans), and paraphrasing at sentence and text levels.

**Evaluated Detectors and Datasets.** In our paper, we evaluate six representative detectors: RADAR (Hu et al., 2023), Detection in the Wild (Wild) (Li et al., 2024), T5Sentinel (Chen et al., 2023), Fast-DetectGPT (Bao et al., 2024), GPTID (Tulchinskii et al., 2024), and LogRank (Mitchell et al., 2023). RADAR, Wild, and T5Sentinel are trained detectors, while Fast-DetectGPT, GPTID, and LogRank are zero-shot detectors. To ensure a fair comparison and assess the detectors' ability to generalize to new data, we carefully select datasets that have not been used in the training or evaluation of these detectors. Table 4 presents an overview of the datasets and domains on which each detector has been evaluated. Several datasets, such as Xsum, SQuAD, and Reddit Writing Prompts, have been used in the evaluation or training of multiple detectors. Although these detectors achieve strong Area Under the Receiver Operating Characteristic (AUROC) scores on these datasets, they do not report the True Positive Rate at a set False Positive Rate (TPR@FPR), which is a crucial metric in real-world scenarios. To address this gap, we aim to evaluate all six detectors on the same datasets using both AUROC and TPR at FPR metrics.

| Method | Datasets |
|---|---|
| RADAR | OpenWebText Corpus (Gokaslan et al., 2019), Xsum (Narayan et al., 2018), SQuAD (Rajpurkar et al., 2016), Reddit Writing Prompts (Fan et al., 2018), and TOEFL (Liang et al., 2023) |
| Wild | Reddit CMV sub-community comments (Tan et al., 2016), Yelp Reviews (Zhang et al., 2015), Xsum (Narayan et al., 2018), TLDR_news[6], ELI5 dataset (Fan et al., 2019), Reddit Writing Prompts (Fan et al., 2018), ROCStories Corpora (Mostafazadeh et al., 2016), HellaSwag (Zellers et al., 2019a), SQuAD (Rajpurkar et al., 2016), and SciGen (Moosavi et al., 2021) |
| T5Sentinel | OpenWebText Corpus (Gokaslan et al., 2019) |
| Fast-DetectGPT | Xsum (Narayan et al., 2018), SQuAD (Rajpurkar et al., 2016), Reddit Writing Prompts (Fan et al., 2018), WMT16 English and German (Bojar et al., 2016), PubMedQA (Jin et al., 2019) |
| GPTID | Wiki40b (Guo et al., 2020), Reddit Writing Prompts (Fan et al., 2018), WikiM (Krishna et al., 2024), StackExchange (Tulchinskii et al., 2024) |
| LogRank | Xsum (Narayan et al., 2018), SQuAD (Rajpurkar et al., 2016), Reddit Writing Prompts (Fan et al., 2018) |

Table 4: Datasets used for training and evaluation by each model. To avoid data leakage and cherry-picking, these datasets are excluded from the current study.

**Comparison to Previous Works.** There are some other papers that have explored similar work to ours, specifically Wang et al. (2024) and Dugan et al. (2024). Our work differs from theirs in some important ways. We do not focus as much on the various methods of red-teaming the detectors in complicated ways. Rather, we explore some more natural methods that an average person might utilize in practice. We also explore in more depth the variability in detector capabilities across various tasks and languages with discussion on potential sources of that difference. And lastly, we utilize newer models, which gives insight into the adaptability of the detectors.

# B  MORE RESULTS AND PROMPTS

This section contains results for detections by models and tasks. It also includes the prompts used in plain prompting.

## B.1  RESULTS BY MODEL

## B.2  PLAIN PROMPTS

Table 12 shows the prompts used for each task in the plain prompting.

| Model | Detector | TPR@.01 | AUROC |
|---|---|---|---|
| **GPT-4o** | Radar | 0.00 | 0.4621 |
| | Fast-DetectGPT | 0.01 | 0.8501 |
| | Wild | 0.01 | 0.4797 |
| | PHD | 0.00 | 0.4809 |
| | LogRank | 0.00 | 0.2774 |
| | T5Sentinel | 0.01 | 0.4930 |
| **Llama-3** | Radar | 0.00 | 0.5501 |
| | Fast-DetectGPT | 0.01 | 0.9089 |
| | Wild | 0.01 | 0.6428 |
| | PHD | 0.00 | 0.7830 |
| | LogRank | 0.00 | 0.5217 |
| | T5Sentinel | 0.01 | 0.7524 |
| **Mistral** | Radar | 0.00 | 0.5387 |
| | Fast-DetectGPT | 0.01 | 0.8628 |
| | Wild | 0.01 | 0.4679 |
| | PHD | 0.00 | 0.8599 |
| | LogRank | 0.00 | 0.4219 |
| | T5Sentinel | 0.01 | 0.8640 |
| **Phi-3** | Radar | 0.00 | 0.4837 |
| | Fast-DetectGPT | 0.01 | 0.0939 |
| | Wild | 0.01 | 0.2640 |
| | PHD | 0.00 | 0.9326 |
| | LogRank | 0.01 | 0.9930 |
| | T5Sentinel | 0.01 | 0.0568 |

Table 5: Code

| Model | Detector | TPR@.01 | AUROC |
|---|---|---|---|
| **GPT-4o** | Radar | 0.03 | 0.3139 |
| | Fast-DetectGPT | 0.47 | 0.9468 |
| | Wild | 0.03 | 0.5078 |
| | PHD | 0.01 | 0.5297 |
| | LogRank | 0.00 | 0.4098 |
| | T5Sentinel | 0.04 | 0.4996 |
| **Llama-3** | Radar | 0.15 | 0.6884 |
| | Fast-DetectGPT | 0.85 | 0.9873 |
| | Wild | 0.08 | 0.6405 |
| | PHD | 0.00 | 0.2532 |
| | LogRank | 0.00 | 0.1682 |
| | T5Sentinel | 0.01 | 0.5269 |
| **Mistral** | Radar | 0.06 | 0.6081 |
| | Fast-DetectGPT | 0.77 | 0.9626 |
| | Wild | 0.01 | 0.5751 |
| | PHD | 0.00 | 0.3283 |
| | LogRank | 0.00 | 0.2540 |
| | T5Sentinel | 0.01 | 0.4767 |
| **Phi-3** | Radar | 0.09 | 0.6485 |
| | Fast-DetectGPT | 0.54 | 0.9143 |
| | Wild | 0.19 | 0.6586 |
| | PHD | 0.00 | 0.2330 |
| | LogRank | 0.00 | 0.1445 |
| | T5Sentinel | 0.01 | 0.3599 |

Table 6: Question Answering

| Model | Detector | TPR@.01 | AUROC |
|---|---|---|---|
| **GPT-4o** | Radar | 0.00 | 0.1771 |
| | Fast-DetectGPT | 0.14 | 0.7731 |
| | Wild | 0.15 | 0.5088 |
| | PHD | 0.00 | 0.4984 |
| | LogRank | 0.00 | 0.2668 |
| | T5Sentinel | 0.03 | 0.5675 |
| **Llama-3** | Radar | 0.01 | 0.5823 |
| | Fast-DetectGPT | 0.23 | 0.7735 |
| | Wild | 0.22 | 0.7099 |
| | PHD | 0.00 | 0.2366 |
| | LogRank | 0.00 | 0.0887 |
| | T5Sentinel | 0.04 | 0.5816 |
| **Mistral** | Radar | 0.00 | 0.3128 |
| | Fast-DetectGPT | 0.09 | 0.5419 |
| | Wild | 0.20 | 0.6657 |
| | PHD | 0.00 | 0.4590 |
| | LogRank | 0.00 | 0.2042 |
| | T5Sentinel | 0.07 | 0.5721 |
| **Phi-3** | Radar | 0.16 | 0.8151 |
| | Fast-DetectGPT | 0.17 | 0.5707 |
| | Wild | 0.70 | 0.9491 |
| | PHD | 0.00 | 0.0785 |
| | LogRank | 0.05 | 0.1095 |
| | T5Sentinel | 0.01 | 0.5638 |

Table 7: Summarization

| Model | Detector | TPR@.01 | AUROC |
|---|---|---|---|
| **GPT-4o** | Radar | 0.05 | 0.6134 |
| | Fast-DetectGPT | 0.69 | 0.9712 |
| | Wild | 0.00 | 0.5466 |
| | PHD | 0.00 | 0.3366 |
| | LogRank | 0.00 | 0.2094 |
| | T5Sentinel | 0.06 | 0.4982 |
| **Llama-3** | Radar | 0.14 | 0.6936 |
| | Fast-DetectGPT | 0.82 | 0.9850 |
| | Wild | 0.00 | 0.6513 |
| | PHD | 0.00 | 0.2364 |
| | LogRank | 0.00 | 0.1562 |
| | T5Sentinel | 0.01 | 0.4590 |
| **Mistral** | Radar | 0.35 | 0.8280 |
| | Fast-DetectGPT | 0.60 | 0.9392 |
| | Wild | 0.00 | 0.5945 |
| | PHD | 0.02 | 0.2799 |
| | LogRank | 0.01 | 0.2483 |
| | T5Sentinel | 0.01 | 0.5036 |
| **Phi-3** | Radar | 0.06 | 0.7626 |
| | Fast-DetectGPT | 0.73 | 0.9005 |
| | Wild | 0.23 | 0.7253 |
| | PHD | 0.00 | 0.1350 |
| | LogRank | 0.00 | 0.0407 |
| | T5Sentinel | 0.00 | 0.2644 |

Table 8: Dialogue

| Model | Detector | TPR@.01 | AUROC |
|---|---|---|---|
| GPT-4o | Radar | 0.00 | 0.2464 |
| | Fast-DetectGPT | 0.46 | 0.9547 |
| | Wild | 0.03 | 0.6328 |
| | PHD | 0.05 | 0.7846 |
| | LogRank | 0.00 | 0.3659 |
| | T5Sentinel | 0.02 | 0.6542 |
| Llama-3 | Radar | 0.43 | 0.8694 |
| | Fast-DetectGPT | 0.92 | 0.9849 |
| | Wild | 0.59 | 0.9388 |
| | PHD | 0.01 | 0.2668 |
| | LogRank | 0.00 | 0.0844 |
| | T5Sentinel | 0.00 | 0.1827 |
| Mistral | Radar | 0.00 | 0.1914 |
| | Fast-DetectGPT | 0.48 | 0.9385 |
| | Wild | 0.04 | 0.5701 |
| | PHD | 0.00 | 0.6644 |
| | LogRank | 0.00 | 0.2980 |
| | T5Sentinel | 0.00 | 0.4285 |
| Phi-3 | Radar | 0.69 | 0.9252 |
| | Fast-DetectGPT | 0.45 | 0.7116 |
| | Wild | 0.26 | 0.9232 |
| | PHD | 0.00 | 0.2642 |
| | LogRank | 0.03 | 0.3134 |
| | T5Sentinel | 0.06 | 0.6680 |

Table 9: Abstract

| Model | Detector | TPR@.01 | AUROC |
|---|---|---|---|
| GPT-4o | Radar | 0.14 | 0.9800 |
| | Fast-DetectGPT | 0.98 | 0.9986 |
| | Wild | 0.00 | 0.9844 |
| | PHD | 0.00 | 0.1159 |
| | LogRank | 0.00 | 0.0109 |
| | T5Sentinel | 0.00 | 0.2526 |
| Llama-3 | Radar | 0.53 | 0.9701 |
| | Fast-DetectGPT | 0.97 | 0.9870 |
| | Wild | 0.44 | 0.9933 |
| | PHD | 0.00 | 0.0230 |
| | LogRank | 0.00 | 0.0061 |
| | T5Sentinel | 0.00 | 0.3243 |
| Mistral | Radar | 0.44 | 0.9830 |
| | Fast-DetectGPT | 1.00 | 0.9990 |
| | Wild | 0.55 | 0.9948 |
| | PHD | 0.00 | 0.0729 |
| | LogRank | 0.00 | 0.0080 |
| | T5Sentinel | 0.00 | 0.2331 |
| Phi-3 | Radar | 0.72 | 0.9100 |
| | Fast-DetectGPT | 0.65 | 0.7642 |
| | Wild | 0.48 | 0.9815 |
| | PHD | 0.00 | 0.0452 |
| | LogRank | 0.10 | 0.1912 |
| | T5Sentinel | 0.01 | 0.4959 |

Table 10: Reviews

| Model | Detector | TPR@.01 | AUROC |
|---|---|---|---|
| **GPT-4o** | Radar | 0.02 | 0.6151 |
| | Fast-DetectGPT | 0.05 | 0.6543 |
| | Wild | 0.11 | 0.5532 |
| | PHD | 0.00 | 0.4469 |
| | LogRank | 0.00 | 0.4135 |
| | T5Sentinel | 0.03 | 0.5792 |
| **Llama-3** | Radar | 0.15 | 0.9057 |
| | Fast-DetectGPT | 0.40 | 0.7819 |
| | Wild | 0.38 | 0.7178 |
| | PHD | 0.00 | 0.2781 |
| | LogRank | 0.00 | 0.2876 |
| | T5Sentinel | 0.02 | 0.4769 |
| **Mistral** | Radar | 0.02 | 0.8374 |
| | Fast-DetectGPT | 0.09 | 0.6568 |
| | Wild | 0.19 | 0.6400 |
| | PHD | 0.00 | 0.3826 |
| | LogRank | 0.01 | 0.4333 |
| | T5Sentinel | 0.03 | 0.5486 |
| **Phi-3** | Radar | 0.18 | 0.9735 |
| | Fast-DetectGPT | 0.06 | 0.5191 |
| | Wild | 0.44 | 0.7872 |
| | PHD | 0.00 | 0.2601 |
| | LogRank | 0.02 | 0.4431 |
| | T5Sentinel | 0.09 | 0.5473 |

Table 11: Translation

| Task | Prompt |
|---|---|
| Code | You are a helpful code assistant that can teach a junior developer how to code. Your language of choice is Python. Don't explain the code, just generate the code block itself. |
| Question Answering | You are a helfpul question answering assistant that will answer a single quesetion as completely as possible given the information in the question. Do NOT using any markdown, bullet, or numbered list formatting. The assistant will use ONLY paragraph formatting. **Respond only in {language}** |
| Summarization | You are a helfpul summarization assistant that will summarize a given article. Provide only the summarization in paragraph formatting. Do not introduce the summary. **Respond in {language}** |
| Dialogue | You are a helpful dialogue generation assistant that will generate a dialogue between people given a short paragraph describing the people involved. Provide only the dialogue. Do not introduce the dialogue. **Respond in {language}** |
| Abstract Writing | You are a helpful abstract writing assistant. You will write an abstract given the content of a paper. Do not provide any other text. You will only provide an abstract. |
| Review Writing | You are a helpful conference paper review assistant. Please provide a detailed review of the following paper, including its strengths, weaknesses, and suggestions for improvement. |
| Translation | You are a helpful translation assistant that will translate a given text into English. Provide only the translation and nothing else. |
| Rewriting | You are a helpful writting assistant. Rewrite the following text to improve clarity and professionalism. Do not provide any other text. Only provide the rewritten text. |

Table 12: The table shows the prompts used in the plain prompting. For GPT, these were used as system prompts, and for huggingface models they were prepended to the questions.

