# OpenReview forum: "An Examination of AI-Generated Text Detectors Across Multiple Domains and Models"
_NeurIPS.cc/2024/Workshop/SafeGenAi — SafeGenAi Poster_

### Official Review · Reviewer_e9YW · 2024-10-09
**Review of AI-Generated Text Detectors Across Multiple Domains and Models**

**Rating:** 6
**Confidence:** 4

**Review:**

The authors evaluate several detectors with their own created datasets through adversarial techniques. Additionally, they demonstrate weaknesses with the AUC metric and propose using the TPR@FPR metric instead.

## Pros
- The authors have done in-depth work into the current state of AI-text detection, as seen in their introduction.
- The datasets cover a wide range of fields that are prone to LLM abuse - from dialogue to translation.
- All attacks/perturbations are described in great detail, which can help reproducibility.

## Feedback
- While the paper acknowledges that it focuses on out-of-distribution (OOD) texts, it would be interesting to see how trained classifiers (on the author's plain prompted texts) perform rather than the pre-trained classifiers. It is not entirely surprising that AI detectors would struggle with OOD texts.  If the detectors struggle with plain prompted OOD texts, they would also naturally struggle with OOD adversarial texts.
- The TPR@FPR metrics are relatively low even for plain-prompting texts. This might diminish the claim that these adversarial methods aren't as effective since any decrease wouldn't be significant.

---

### Official Review · Reviewer_QQhm · 2024-10-09
**Need more database background and experiment process**

**Rating:** 6
**Confidence:** 5

**Review:**

The paper provides a valuable evaluation of AI-generated text detectors across multiple domains and models. It highlights the limitations of using AUROC as a metric and suggests using TPR@FPR as a more reliable indicator of a detector's effectiveness.

Cons:

1. Can you provide more information on the selection process for the datasets used in each domain? How did you ensure that the datasets represent new, unseen data and prevent data leakage?
2. In the red teaming experiments, did you consider any other attack methods besides PAIR and paraphrasing? If not, why was that the case?
3. Have you considered the potential impact of model size and training data size on the performance of the detectors? Could these factors contribute to the observed limitations of the detectors?
4. How did you determine the TPR@FPR thresholds for evaluating the detectors? Were these thresholds chosen based on previous research or empirical observations?
5. Could you provide a more thorough discussion of related work and background in the revised version of the paper?
6. Have you considered the potential limitations of the detectors themselves, such as their reliance on certain features or the possibility of adversarial attacks targeting the detectors? How do these limitations affect the overall performance of the detectors?

---

### Official Review · Reviewer_M54G · 2024-10-10

**Rating:** 7
**Confidence:** 4

**Review:**

**Summary**

This paper evaluates the effectiveness of four open-source AI text detectors (RADAR, Wild, Fast-DetectGPT, and GPTID) across seven different domains of text. Generations from multiple different LLMs are evaluated under both plain and adversarial prompting scenarios. The results show that the AUROC metric used to evaluate classifiers in existing work is poorly suited to this task, as it fails to characterize detection accuracy at the low level of false positives required for practical applications.

**Strengths**

- The breadth of experimental results is significant and provides interesting insights into how the performance of existing detection methods varies significantly across different domains of text
- Multiple different adversarial prompts are investigated, which do not produce significantly different detection results but are interesting nonetheless
- The finding that AUROC is a poor predictor of detection accuracy at low false positive rates is valuable, and the authors correctly point out that TPR at a low FPR (e.g., 0.01) is most important for practical applications of AI text detectors

**Weaknesses**

- The findings of the analysis conducted in this paper are valuable, but the technical contribution of this work is somewhat limited as it does not propose any new methods for this problem.
- No commercial APIs for AI text detection are evaluated as part of this study. It would have been interesting to see how such solutions compare in performance to open-source text detectors across the different domains evaluated in this study.
- The main paper is missing basic details such as a description of the evaluated text detection methods. While this is provided in the appendix, it would be more helpful to describe the models before results are discussed in the main paper.
- Some details in Section 3 are unclear. For example, why does a dataset imbalance lead to a more accurate estimation of TPR@FPR (lines 208-210)? How can you calculate TPR at different values of FPR without using a threshold to control the sensitivity of the classifier (lines 239-240)?